# The Effects of Carbohydrate versus Fat Restriction on Lipid Profiles in Highly Trained, Recreational Distance Runners: A Randomized, Cross-Over Trial

**DOI:** 10.3390/nu14061135

**Published:** 2022-03-08

**Authors:** Alex Buga, Gary L. Welton, Katie E. Scott, Adam D. Atwell, Sarah J. Haley, Noah J. Esbenshade, Jacqueline Abraham, Jeffrey D. Buxton, Dana L. Ault, Amy S. Raabe, Timothy D. Noakes, Parker N. Hyde, Jeff S. Volek, Philip J. Prins

**Affiliations:** 1Department of Human Sciences, The Ohio State University, Columbus, OH 43210, USA; buga.1@osu.edu (A.B.); volek.1@osu.edu (J.S.V.); 2Department of Exercise Science, Grove City College, Grove City, PA 16127, USA; weltongl@gcc.edu (G.L.W.); scottke1@gcc.edu (K.E.S.); adam.atwell68@gmail.com (A.D.A.); haleysj1@gcc.edu (S.J.H.); esbenshadenj1@gcc.edu (N.J.E.); jacquelineabraham77@gmail.com (J.A.); buxtonjd@gcc.edu (J.D.B.); aultdl@gcc.edu (D.L.A.); 3Department of Human Ecology, Youngstown State University, Youngstown, OH 44555, USA; asraabe@ysu.edu; 4Department of Applied Design, Cape Peninsula University of Technology, Cape Town 8000, South Africa; tim.noakes0207@gmail.com; 5Department of Kinesiology, University of Northern Georgia, Dahlonega, GA 30597, USA; parker.hyde@ung.edu

**Keywords:** low-carbohydrate, low-fat, crossover, endurance athletes, cholesterol, lipid profiles

## Abstract

A growing number of endurance athletes have considered switching from a traditional high-carbohydrate/low-fat (HCLF) to a low-carbohydrate/high-fat (LCHF) eating pattern for health and performance reasons. However, few studies have examined how LCHF diets affect blood lipid profiles in highly-trained runners. In a randomized and counterbalanced, cross-over design, athletes (*n* = 7 men; VO_2max_: 61.9 ± 6.1 mL/kg/min) completed six weeks of two, ad libitum, LCHF (6/69/25% en carbohydrate/fat/protein) and HCLF (57/28/15% en carbohydrate/fat/protein) diets, separated by a two-week washout. Plasma was collected on days 4, 14, 28, and 42 during each condition and analyzed for: triglycerides (TG), LDL-C, HDL-C, total cholesterol (TC), VLDL, fasting glucose, and glycated hemoglobin (HbA1c). Capillary blood beta-hydroxybutyrate (BHB) was monitored during LCHF as a measure of ketosis. LCHF lowered plasma TG, VLDL, and TG/HDL-C (all *p* < 0.01). LCHF increased plasma TC, LDL-C, HDL-C, and TC/HDL-C (all *p* < 0.05). Plasma glucose and HbA1c were unaffected. Capillary BHB was modestly elevated throughout the LCHF condition (0.5 ± 0.05 mmol/L). Healthy, well-trained, normocholesterolemic runners consuming a LCHF diet demonstrated elevated circulating LDL-C and HDL-C concentrations, while concomitantly decreasing TG, VLDL, and TG/HDL-C ratio. The underlying mechanisms and implications of these adaptive responses in cholesterol should be explored.

## 1. Introduction

Endurance athletes are encouraged to consume a carbohydrate-rich diet (7–12 g/kg/day) to achieve peak performance [1,2]. Carbohydrate-restriction (<50 g/day) has been adopted in recent years by an increasing number of athletes who want to test its putative roles in training, health, weight loss, body composition, gastrointestinal tolerance, and recovery [3,4]. The diet relies on the main premises that (1) an increase fat oxidation to a threshold where lipids become the predominant fuel during prolonged, submaximal exertion (~65% VO_2max_) confers a substrate advantage that does not need refueling [5,6,7] and (2) maintaining this elevated fat oxidation rate may delay glycogen degradation for race stages where high-intensity work (>70% VO_2max_) output is required (i.e., sprints) [4,5,8].

The evidence supporting low-carbohydrate/high-fat (LCHF) diets for aerobic performance is mixed, and whether one dietary strategy is best suited for athletic activities is still being extensively investigated. However, a well-documented effect when switching from a habitual diet to an LCHF diet is observed in blood lipids. Markers such as total cholesterol (TC), low/high-density lipoproteins (LDL/HDL) and triglycerides (TG), expressed either as absolute values or relative ratios, are measured to characterize cardiovascular risk. The goal of lifestyle interventions is to primarily lower TC, LDL, and TGs and increase HDL to induce a favorable lipid profile [9,10], an effect that LCHF interventions have demonstrated in non-athletic populations undergoing weight-loss [11,12]. Studies that previously tested LCHF responses in athletes reported variable results, with some documenting a decrease in circulating lipids [13], whereas others reported neutral [14,15] or an increasing trend [16,17] across all lipid parameters. Additionally, there is strong evidence to suggest that aerobic exercise alone may sufficiently act as a positive modulator of serum lipids, primarily HDL, independent of other factors [18,19]. Despite the uses of LCHF in a range of sports [20,21,22], few studies have carefully examined how diet affects lipid profiles in endurance athletes [23].

In a cross-sectional study of ultra-endurance runners, the group habitually consuming a LCHF diet (>20 months; *n* = 11) demonstrated greater circulating cholesterol, both as total, large diameter LDL, and HDL, compared to the high-CHO comparison group [24,25]. Lambert et al. conducted a small crossover study in cyclists (*n* = 5) and showed that LCHF did not significantly alter blood lipids whilst preserving the cycling performance compared to a habitual mixed diet [26]. Lastly, O’Neal et al. revealed that, in well-trained middle-aged runners, three weeks of a mixed diet crossed over to LCHF (*n* = 8), raised TC and LDL concentrations (~30 and 20 mg/dL increase, respectively), but not HDL, while TG decreased significantly, independent of diet [27]. These findings suggest that blood lipid profiles in endurance populations may vary in response to both diet and exercise (i.e., LDL/HDL increase; TG decrease). The most plausible explanation for between-group variability is methodology differences across studies. Our previously published cross-over approach [15] was intended to evaluate both within- and between-group changes and has been recently recommended in a LCHF diet and performance review paper [23] as a model for future studies that plan to evaluate diet-dependent effects in athletes.

The main goal of this exploratory study was to determine the effects of LCHF or HCLF feeding on fasted blood lipids and glucose markers in long distance runners. To answer this question, we performed the first randomized and balanced, cross-over trial in highly trained endurance athletes (*n* = 7 men). Each athlete was guided on how to implement LCHF and HCLF lifestyles ad libitum for six weeks, separated by a two-week washout between experimental conditions. Dietary compliance during the LCHF condition was evaluated using food logs and tracking the primary circulating ketone body in capillary blood (beta-hydroxybutyrate; BHB). Plasma was analyzed for total cholesterol (TC), very-low/low/high-density lipoproteins (VLDL, LDL-C, HDL-C), triglycerides (TG), glucose and glycated hemoglobin (HbA1C).

## 2. Materials and Methods

### 2.1. Experimental Design and Participants

This was an exploratory, randomized, counterbalanced crossover-trial study design. The main goal was to evaluate the blood lipid responses in highly trained, recreational endurance athletes who were motivated to undergo 12 weeks of experimental, dietary intervention phases concurrent with their training regimen. Inclusion criteria were divided into three objective categories: running performance (<21′00″ 5-km within 3 months of study enrollment; >32 km of running per week; >2 years of running experience); age (18–45 years); habitual dietary intake (>50% total energy needs from carbohydrates). Exclusion criteria included hypercholesterolemia (>200 mg/dL), hypertriglyceridemia (>150 mg/dL), habitually consuming a ketogenic or low-carbohydrate diet (<20% total energy needs from carbohydrates) or being prescribed lipid- or glucose-lowering medication. 

Before enrolling in the study, participants were fully informed of any associated risks and discomforts prior to giving their written informed consent to participate. The experimental protocol was approved by the Grove City College Institutional Review Board prior to implementation. Participants (*n* = 7 men) who met the criteria were invited to in-person consent visit where the protocol and study responsibilities were described in greater detail. Each participant received a unique study identification number to protect their real identity. The experimental order was established by codifying each participant (1–7) and dietary conditions (A = LCHF; B = HCLF) into a randomized, online number generator (www.randomizer.org; accessed on 19 June 2018). After randomization, each participant was assigned to an ad libitum, low-carbohydrate/high-fat diet (LCHF) or a high-carbohydrate/low-fat diet (HCLF) for six weeks. A two-week, also ad libitum, mixed diet (i.e., CHO > 50 g/day and >20% fat) washout stage separated the two experimental dietary phases. Participants reported to the testing site bi-weekly for data collection. Baseline characteristics (Table 1) and a graphical summary of experimental design are presented below (Figure 1).

### 2.2. Nutrition and Exercise Guidelines

A registered dietitian educated and guided each athlete a priori of experimental phases on how to implement LCHF and HCLF guidelines at home through direct counseling and handouts. The primary macronutrient targets for LCHF and HCLF were expressed both as percentage of total daily energy intake (% en) and per gram basis: LCHF: <50 g/day carbohydrate, 75–80% en fat, 15–20% en protein.HCLF: 60–65% en carbohydrate, 20% en fat, 15–20% en protein.

Participants were explicitly instructed to consume the diets until they reach satiety. Consumption of a wide range of foods was encouraged to minimize micronutrient deficiencies. To ensure that mineral status was met for LCHF, we recommended including an additional 1–2 g/day of iodized table salt to offset the additional loss of sodium associated with a reduction in total carbohydrate intake [28]. Weekly energy intake and relative macronutrient distribution was monitored and estimated via 3-day weighed food records, capturing two consecutive weekdays and a weekend day. A digital scale (Ozeri ZK14-S Pronto, San Diego, CA, USA) calibrated to the nearest ±0.1 g was provided to each athlete prior to experimental phases to improve food tracking accuracy (intended for both dry and cooked items). Dietary macro- and micronutrients were calculated by the same registered dietitian using advanced nutrient software (Nutritionist Pro, Axxya Systems, Redmond, WA, USA). In addition to food records, compliance to the LCHF dietary regimen was monitored by daily, morning capillary blood ketone measurements as determined by beta-hydroxybutyrate (BHB).

To minimize confounding exercise effects, participants were instructed to select and maintain a constant training intensity and volume that they could adhere to for 14 weeks. Subjects were instructed to record their training habits (mode/duration/intensity) one week prior to commencing experimental dietary phases. Post-hoc training load analysis revealed no differences in free-living exercise habits throughout the study. Detailed description of exercise testing during the study was reported previously [15].

### 2.3. Laboratory Protocols

The data presented in this manuscript was collected as part of a larger project [15] examining the effects of LCHF and HCLF on dietary and exercise adherence, running performance, physiological and metabolic adaptations, and change in body composition. Detailed description of general methods and other results have been previously detailed by Prins et al. [15]. In brief, a sequence of tests was performed on day 4, 14, 28, and 42 during each dietary phase. Participants reported to the testing laboratory (Grove City College Exercise Science Human Performance Laboratory) between 6:00 and 9:00 a.m. after an overnight fast (8–12 h). Upon arrival, hydration status was assessed in urine via specific gravity (USG < 1.020) using a portable light refractometer (Reichert™, Buffalo, NY, USA). Athletes who did not meet this threshold were offered ~250 mL of water and re-tested for hydration.

Following an 8–12-h fast, a trained phlebotomist collected fasting venous blood samples via venipuncture. Before sampling, participants were asked by a researcher if they had anything to eat or drink that morning. Venous samples were taken between 6 a.m. and 9 a.m. Samples were collected in a lithium heparin BD Vacutainer Plasma Tube and immediately placed on ice after collection. Whole blood was analyzed within 10 min of collection using Abaxis Piccolo Xpress point of care chemistry analyzer (Abaxis Inc., Union City, CA, USA) and Lipid Panel Plus discs assays for the measurement of total cholesterol (TC), low density lipoprotein cholesterol (LDL-C), high density lipoprotein cholesterol (HDL-C), triglycerides (TG), very low-density lipoprotein (VLDL), TC to HDL-C ratio (TC/HDL-C), TG to HDL-C ratio (TG/HDL-C) and glucose. LDL-C was calculated automatically using the Friedwald equation [29]. Prior to blood analysis, calibration and quality control measures were taken by running control reagents, and all values fell within the acceptable device range. Glycated hemoglobin was analyzed using an Alere Afinion HbA1c Analyzer (Alere Technologies).

Fasted capillary BHB was measured by participants every morning via finger sticks (<100 µL blood sample) using a commercially available glucometer fitted for ketone reagent strips (Precision Xtra, Abbott Diabetes Care Inc., Almeda, CA, USA). Daily BHB values were retrieved on test days from the internal device memory and manually stored into a secure database by the laboratory team members.

### 2.4. Statistics

Analyses were performed using SPSS ver. 25 (SPSS, Inc., Chicago, IL, USA). Two-tail *α* significance was set at *p* < 0.05. To address our exploratory objective, we analyzed main effects and interactions between LCHF and HCLF using a 2 (condition) × 4 (time) repeated measures ANOVA. Subject characteristics at baseline are presented as descriptive variables. All the variables of interest analyzed were screened for normality using Shapiro-Wilks test. Assumption of sphericity was confirmed using Mauchly’s test; variables that violated sphericity were treated with the Greenhouse-Geiser correction. Bonferroni correction was applied for multiple post-hoc comparisons. All data is presented as mean ± SD. Figures were created with BioRender^®^ and graphs were designed in GraphPad Prism (GraphPad Software, Inc., San Diego, CA, USA; ver. 9.1). 

## 3. Results

Based on food records, participants adhered to the general principles of both diet guidelines (Table 2). There were no significant differences in average total energy intake between the two conditions (ΔLCHF − HCLF = 110 kcal/day; *p* = 0.686). Participants met most macronutrient goals; during the LCHF experimental condition participants reported weekly carbohydrate intake below the <50 g/d threshold (43 ± 6 g/day; 6% en), whereas average CHO intake during HCLF increased nearly ~10-fold (402 ± 32 g/day; 56% en). Daily fasted capillary blood βHB concentrations averaged 0.5 ± 0.05 mmol/L throughout the six-week LCHF condition (Figure 2). A subset of participants met the nutritional ketosis threshold of BHB > 0.5 mmol/L (*n* = 2), whereas others met the borderline criteria (*n* = 1) or remained below this level (*n* = 4). Weekly BHB averages did not change significantly over time relative to the first week of the LCHF condition (*p* = 0.286).

Dietary fat intake increased during LCHF condition and provided nearly ~2.5 times more relative energy than HCLF (69 vs. 28% en). Consequently, LCHF consumed more grams of cholesterol, saturated, monounsaturated, and polyunsaturated fat, and omega-3 fatty acids (EPA/DHA) than HCLF (*p* < 0.05). LCHF consumed more dietary protein, both expressed as a total relative energy percentage (25 vs. 15% en; *p* < 0.001) and per kilogram of bodyweight ratio (2.68 vs. 1.55 g/kg; *p* < 0.001). Additionally, LCHF consumed approximately 7× more sugar (132 vs. 18 g; *p* < 0.001) and three times more dietary fiber (30 vs. 10 g; *p* < 0.001) than LCHF (Figure 3).

The lipid panel revealed robust changes across all plasma markers (Table 3). When comparing the LCHF diet to HCLF (mean ± SD), the LCHF increased: TC (197 ± 17 vs. 153 ± 20 mg/dL; Δ = 25%; *p* = 0.001), LDL-C (108 ± 17 vs. 74 ± 13 mg/dL; Δ = 38%; *p* = 0.001), and HDL-C (71 ± 17 vs. 61 ± 16 mg/dL; Δ = 15%; *p* = 0.015). LCHF also decreased TG (74 ± 7 vs. 97 ± 14 mg/dL; Δ = −27%; *p* = 0.005), VLDL (15 ± 2 vs. 19 ± 3 mg/dL; Δ = −26%; *p* = 0.004), and TG/HDL-C ratio (1.1 ± 0.3 vs. 1.8 ± 0.6; Δ = −44%; *p* = 0.001) (Figure 4). 

There were no significant changes detected in the glucose panel. LCHF exhibited non-significantly lower fasted plasma glucose (83.3 ± 4.0 vs. 88.7 ± 8.6 mg/dL; Δ = −6.3%; *p* = 0.107) and HbA1c (5.0 ± 0.1 vs. 5.0 ± 0.2 %; Δ = −0.2%; *p* = 0.821) relative to HCLF throughout the experimental phases.

## 4. Discussion

The results presented herein were collected as part of the first exploratory cross-over study that measured LCHF versus HCLF diet effects in highly trained endurance athletes [15]. Overall, there were no detriments to exercise performance during either experimental condition. Both HCLF and LCHF interventions were adequately energy-matched (~42 kcal/kg/day) and confirmed by the stable bodyweight. Large between-group differences were detected in macronutrients, essential fatty acids, fiber, and sugar content over the course of six weeks. Dietary adherence to the LCHF protocol was confirmed by daily fasted capillary blood BHB. Significant diet effects were detected in key lipid panel markers (TC, HDL, LDL, VLDL) that increased during LCHF compared to HCLF, whereas TGs remained moderately lower throughout the LCHF intervention. HDL, VLDL and TG did not change over time. Neither intervention influenced glucose or HbA1C meaningfully. 

We expected the LCHF intervention to increase circulating BHB within a predicted nutritional ketosis range (0.5–4.0 mM), Studies and reviews of well-formulated KD suggest that a protein range between 1.2–1.5 g/kg bodyweight to spare lean body mass losses while concurrently promoting BHB maintenance within nutritional ketosis range [12,30]. Phinney et al. [5] provided recreational and competitive cyclists with 1.75 g protein/kg/day and detected a 2.2 mmol/L serum BHB average for 4 weeks. LaFountain et al. [14] established the protein goals between 0.6–1.0 g/kg/day and advised the military cohort to adjust their dietary macronutrient intake on a weekly basis, resulting in a mean 1.2 mM capillary blood BHB over 12 weeks. Our participants reported a protein intake closer to 2.6 g/kg day, and higher than reported in the carbohydrate-restricted literature. Items commonly regarded as “low-carbohydrate” provide high-biological value protein (i.e., meats, eggs, dairy) that may blunt BHB responses due to their intrinsic insulinogenic and gluconeogenic properties [31]. It is plausible that the involuntary overconsumption of these food items after switching from a habitual diet to a LCHF diet may explain low circulating BHB, although unsurprising, even in carefully measured free-living conditions. If ketonemia is the primary goal, future studies should explore ways to manipulate macronutrients based on capillary blood measurements to maintain a desired BHB target.

According to the normative thresholds for dyslipidemia and cardiovascular risk [9,10,32,33], the HCLF lipid panel was within range for TC (<190–200 mg/dL), LDL (<100 mg/dL), HDL (>40–45 mg/dL) and TG (<150 mg/dL), whereas LCHF was marginally above range for TC and LDL. The peak TC and LDL concentration detected at six-weeks during LCHF fits the National Cholesterol Education Program criteria of borderline high (TC: 200–239 mg/dL) and near or above optimal (LDL: 100–129 mg/dL) [32]. These results were anticipated for several reasons. Firstly, LCHF interventions have previously shown to increase serum lipids in both sedentary [34,35,36] and trained populations [5,13,16,24,25,27], albeit not always in athletes [20,21]. In our case, highly trained endurance athletes did experience a significant change over time in TC and LDL that was significant from day 14 and thereafter. Secondly, the diets were designed to be isoenergetic to avoid confounding weight loss effects. Volek et al. published a review of 16 clinical trials comparing changes in lipids between LCHF and HCLF and discovered that the greatest percent change in TC and LDL was in the only isoenergetic trial [37]. In other words, weight loss exerts a large, positive effect on circulating lipids, which makes diet-dependent effects difficult to single out from other factors. Because all participants were weight-stable, the change in circulating lipids from HCLF to LCHF suggests that this change needed to occur for it to accommodate 137 g (+254%) of extra dietary fat. 

Hypercholesterolemic responses in trained populations adopting a LCHF may be partly explained by BHB turnover. Sedentary individuals can produce between 50–100 g of BHB per day (4 kcal/g BHB) depending on insulin sensitivity and diet composition [38]; however, athletes may detect lower circulating BHB values because their circulating ketones are (1) shunted towards energy production or (2) indirectly lost during ketogenesis due to elevated work rate. Kackley et al. demonstrated that athletes in ketosis experience a decrease in BHB when cycling to exhaustion [39] while other authors reported no change or a slight increase in BHB after acute exercise [8,40]. Recently, Creighton et al. [25] speculated that chronic, high-energy expenditure may be a causal factor in lowering BHB in ultramarathon runners explained in part by ketones crossing from the mitochondrial space into the cytosol. In brief, the authors hypothesized that acetoacetate, a precursor of BHB, may often exit the mitochondrial space and convert in the cytosol to acetoacetyl-CoA, thus participating in TCA anaplerosis as acetyl-CoA. Such ketone losses provide the bridge necessary to initiate cholesterol synthesis via cytosol-dependent HMG-CoA reductase action, explaining a previously described phenomenon characterized as “hypercholesterolemia paradox” [12,25]. Altogether, these mechanisms merit further examination to understand if ketosis can explain circulating cholesterol responses in athletic populations.

LCHF intervention provided more dietary fat from cholesterol, SFA/MUFA/PUFA and EPA/DHA than HCLF. Dietary cholesterol is overall a poor surrogate of serum cholesterol in healthy individuals due to homeostatic feedback inhibition on endogenous cholesterol synthesis [41]. Fatty acid saturation, however, particularly SFA, has shown to have the largest influence on TC and LDL (including small density LDL) species, and is used more reliably to describe effects on blood cholesterol [34,37,42]. Although LDL particle size was not included in our analysis goals, the literature indicates that ketogenic diets may favor large, “buoyant” LDL species over smaller, pro-atherogenic “dense” LDL that correlate with cardiovascular injury [43], whereby HCLF may not promote buoyant LDL subclasses [34,42]. The plateau in TC and LDL concentrations at day 28 and thereafter suggests that no further increase was expected in these lipid parameters. Despite the observed plateau it would be prudent to continue measuring blood lipids, regardless of health and activity status, by ordering an extensive lipid panel that includes variables such as particle size, apolipoproteins species (A/B), Lp(a), and more categories for comprehensive risk factor assessment [33].

A notable outcome of LCHF was its ability to lower TGs by ~20 mg/dL (−24%) and increase HDL by ~10 mg/dL (+17%) compared to HCLF. The significant condition and time effects were reflected in LCHF ability to robustly influence TG and HDL from day 4 and thereafter, and beyond HCLF intervention. LCHF have been consistently documented to positively affect TG and HDL as a result of increased lipid fuel utilization as determined by carbohydrate-restriction [3,12,37], with [37] and without weight-loss [42], MUFA/EPA/DHA intake [37,44,45], and exercise [25]. In recent years TG/HDL and TC/HDL ratios below <2.5 and <5.0, respectively, have been included in cardiovascular research as screening prediction models for cardiovascular risk assessment to expand on the interpretation of single parameter lipid panel results [46,47]. In our study LCHF decreased the TG/HDL ratio from 1.025 to 0.97 (−6%), and increased TC/HDL from 2.5 to 3.0 (+20%), with both ratios tracking the TG and TC trends closely rather than HDL per se. From a prospective, cardiovascular risk standpoint there is no clear sign that this population would have reached an unwarranted threshold described in the literature. Studies longer than six weeks in duration may provide more insight into how these lipid parameters evolve longitudinally, albeit questionably within the results of this study, considering that TC and TG lacked significant changes from day 28 and thereafter during the LCHF diet.

The carbohydrate-restricted condition elicited a non-significant reduction (−9%) in circulating blood glucose compared to dietary fat restriction. This result was somewhat expected based on the prescribed diet and health of participants prior to enrolment. All participants had normal glucose metabolism as evidenced by normal fasting blood glucose (<100 mg/dL) and glycosylated hemoglobin (≤5.6%), thus the absolute change from baseline was likely too small to produce a significant effect. Carbohydrate restriction has shown significant effects on elevated fasting glucose in metabolically impaired individuals, namely in individuals diagnosed with type II diabetes [48] and metabolic syndrome [42], a magnitude of change large enough to reach significance relative to the degree of carbohydrate restriction [49]. Although rare, fasting glucose impairment may sometimes occur in recreationally elite-athletes, as it has been previously identified with the use of continuous glucose-monitoring [50]. Irrespective of training status (sedentary; amateur; professional) diet is strongly associated with metabolic health, and athletes should consider talking with a registered dietitian about the best dietary practices for their sport before making changes to their habitual feeding regimen.

## 5. Conclusions

Healthy, well-trained male distance runners 18–45 years of age demonstrated elevated circulating cholesterol, primarily as LDL-C and HDL-C, in response to a six-week *ad libitum* LCHF diet relative to a HCLF diet. The LCHF diet also decreased TG. As a first exploratory study, we were able to report significant diet-dependent effects in most markers of interest, highlighting merit in conducting further testing. Future studies focused on LCHF diets in athletes are encouraged to use advanced lipid quantitation methods (i.e., NMR), longer study durations, and frequent performance assessments to answer novel questions about the impact of CHO-restriction on metabolic health and running performance.

## Figures and Tables

**Figure 1 nutrients-14-01135-f001:**
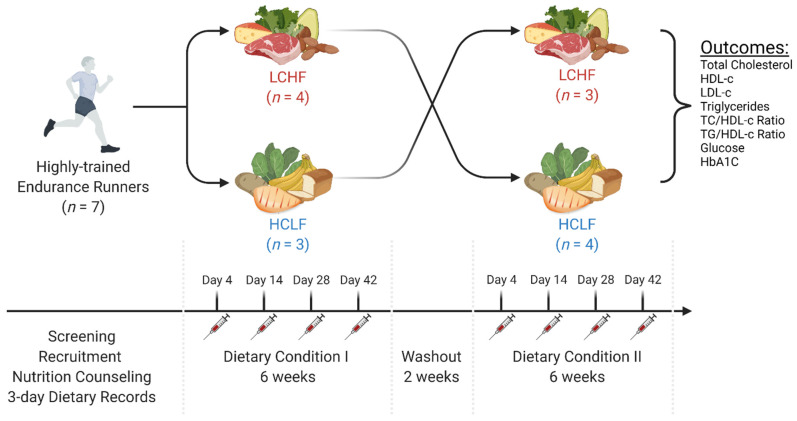
Experimental Approach. In a randomized, counterbalanced, cross-over manner, seven recreationally trained endurance athletes were assigned to consume either a low-carbohydrate/high-fat diet (LCHF) or high-carbohydrate/low-fat diet (HCLF) for six weeks. A two-week washout was allowed between each dietary phase. All participants maintained a constant exercise regimen and a 3-day food log throughout the experimental phases. Repeat intravenous and capillary blood measures were collected on days 4, 14, 28, and 42 during each feeding condition. The exploratory goal of this study was to assess the main effects and interactions of LCHF and HCLF on comprehensive blood lipid and glucose panel markers. This figure was created with BioRender^®^.

**Figure 2 nutrients-14-01135-f002:**
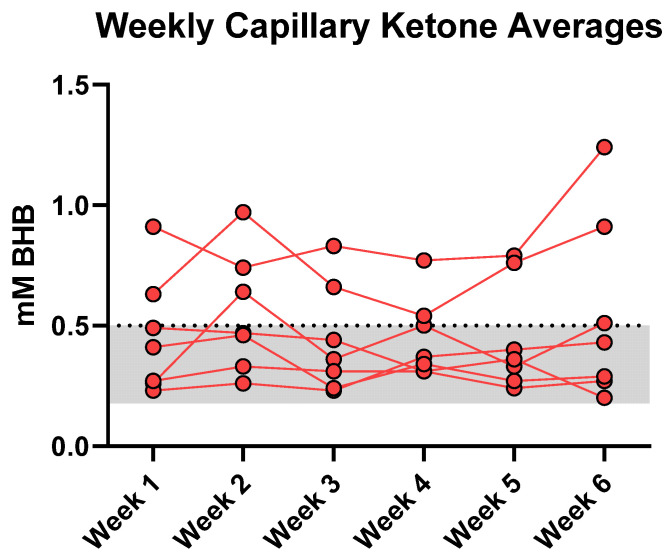
Fasted Capillary Ketone Responses. Each data point denotes a 7-day BHB average. The LCHF feeding phase modestly elevated circulating fasted BHB into the range of nutritional ketosis. Average weekly BHB plateaued at 0.5 ± 0.05 mmol/L (range: 0.20–1.24 mmol/L) over the six weeks. A 1 (condition) × 6 (time) RM ANOVA revealed that BHB concentrations remained stable over time from WK1 and thereafter (*p* = 0.29).

**Figure 3 nutrients-14-01135-f003:**
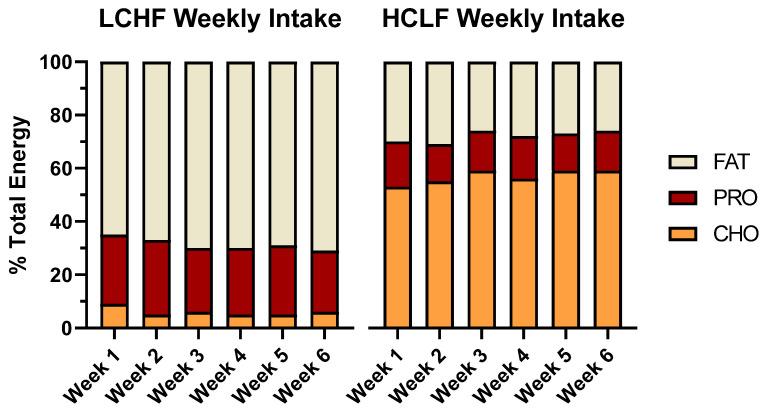
Mean participant energy intake and macronutrient distribution. The 3-day dietary records collected from all participants (*n* = 7) are presented as a weekly average of percent energy distribution from dietary fat, protein, and carbohydrates. A one-way ANOVA between diet conditions revealed significant differences across all relative macronutrient categories (*p* < 0.001), but no significant differences in total energy intake (*p* > 0.05).

**Figure 4 nutrients-14-01135-f004:**
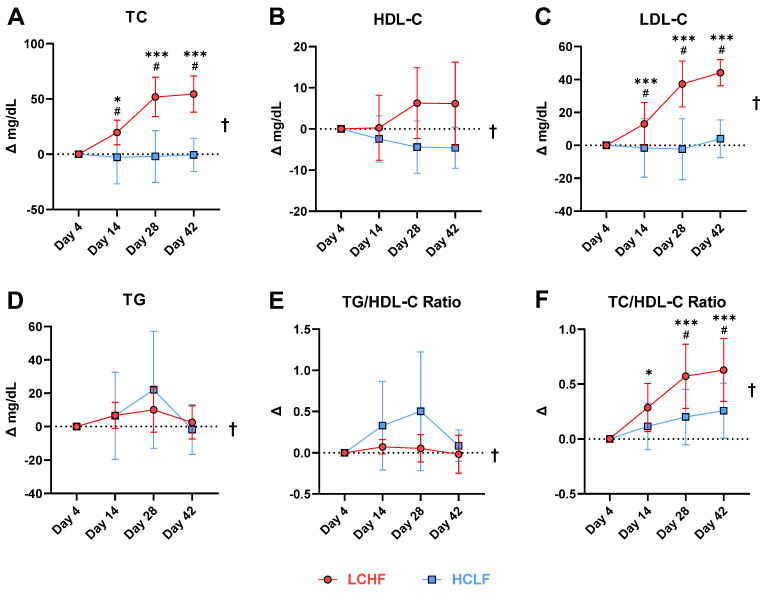
Baseline normalized change in plasma lipid markers. (**A**) total cholesterol, (**B**) HDL-C cholesterol, (**C**) LDL-C cholesterol, (**D**) triglycerides, (**E**) triglyceride to HDL-C ratio, and (**F**) total cholesterol to HDL-C cholesterol ratio. ANOVA revealed a main effect of diet in all the markers analyzed († = *p* < 0.05). Overall, LCHF diet lowered circulating triglycerides and TG/HDL-C ratio while simultaneously increasing HDL-C when compared to the HCLF diet. A significant interaction was detected for total cholesterol, LDL-C, and TC/HDL-C ratio. Compared to HCLF, the LCHF diet elicited significantly higher cholesterol and LDL-C values, and greater TC/HDL-C ratios at day 14 and thereafter. Main effects: Time: *, *** *p* < 0.05/0.001 compared to Day 4; Condition effects: † = *p* < 0.05; Interaction: # = *p* < 0.05 between conditions at indicated timepoints. Data presented as mean ± SD; *n* = 7.

**Table 1 nutrients-14-01135-t001:** Subject Characteristics (*n* = 7).

Variable	Mean	±	SD
Age (years)	35.6	±	8.4
Height (cm)	178.7	±	4.1
Weight (kg)	68.6	±	1.6
BMI (kg/m^2^)	21.5	±	1.1
Body Fat (%) *	5.0	±	1.3
Fat Mass (kg)	3.5	±	1.0
Lean Mass (kg)	65.1	±	1.5
VO_2max_ (mL/kg/min)	61.9	±	6.1
Running Distance Per Week (km)	63.0	±	27.1
Running Experience (years)	15.1	±	7.1

* Determined by bioelectrical impedance. VO_2max_, maximal oxygen consumption.

**Table 2 nutrients-14-01135-t002:** Daily Dietary Composition in the Low-Fat and Low-Carbohydrate Diet Groups Over the Course of the Study.

Variable	Week 1		Week 2		Week 3		Week 4		Week 5		Week 6		Overall Mean		*p*-Value
	LCHF	HCLF	LCHF	HCLF	LCHF	HCLF	LCHF	HCLF	LCHF	HCLF	LCHF	HCLF	LCHF	HCLF	
Energy (Kcal/day)	2574 ± 455	2944 ± 943	3324 ± 543	3118 ± 872	2905 ± 400	2496 ± 661	3243 ± 1132	2886 ± 1094	2828 ± 382	3014 ± 1035	2808 ± 1011	2563 ± 507	2947 ± 284	2837 ± 251	0.686
Carbohydrate (g)	52 ± 16	396 ± 174	44 ± 16	427 ± 132	47 ± 11	371 ± 118	40 ± 10	385 ± 113	36 ± 13	453 ± 211	39 ± 18	379 ± 86	43 ± 6	402 ± 32	0.001
Protein (g)	161 ± 34	118 ± 31	228 ± 77	107 ± 27	170 ± 31	94 ± 20	204 ± 104	112 ± 32	182 ± 49	106 ± 34	157 ± 39	99 ± 28	184 ± 28	106 ± 9	0.001
Fat (g)	191 ± 63	99 ± 36	247 ± 44	110 ± 37	226 ± 41	73 ± 16	250 ± 73	88 ± 38	216 ± 33	89 ± 36	227 ± 98	75 ± 30	226 ± 21	89 ± 14	0.001
Carbohydrate (%)	9 ± 3	53 ± 10	5 ± 2	55 ± 6	7 ± 1	58 ± 4	5 ± 2	56 ± 4	5 ± 2	59 ± 10	5 ± 2	59 ± 9	6 ± 1	56 ± 3	0.001
Protein (%)	26 ± 8	17 ± 5	27 ± 7	14 ± 2	24 ± 5	15 ± 2	25 ± 4	16 ± 3	26 ± 5	14 ± 4	23 ± 5	15 ± 4	25 ± 2	15 ± 1	0.001
Fat (%)	65 ± 11	30 ± 7	67 ± 6	31 ± 5	70 ± 4	26 ± 2	70 ± 3	28 ± 3	68 ± 4	27 ± 9	71 ± 4	26 ± 6	69 ± 2	28 ± 2	0.001
Cholesterol (mg)	938 ± 234	306 ± 173	1283 ± 233	248 ± 127	1033 ± 96	229 ± 104	1365 ± 410	293 ± 227	1317 ± 191	260 ± 147	1261 ± 376	249 ± 188	1199 ± 127	264 ± 30	0.001
Saturated fat (g)	63 ± 21	33 ± 13	97 ± 28	36 ± 16	90 ± 32	22 ± 4	100 ± 28	28 ± 13	90 ± 12	29 ± 17	95 ± 42	24 ± 12	89 ± 13	29 ± 5	0.001
Monounsaturated fat (g)	64 ± 28	28 ± 12	82 ± 18	33 ± 12	76 ± 14	22 ± 6	74 ± 20	26 ± 15	68 ± 19	22 ± 13	73 ± 31	23 ± 12	73 ± 6	26 ± 5	0.001
Polyunsaturated fat (g)	21 ± 9	14 ± 6	24 ± 7	18 ± 5	24 ± 10	15 ± 7	23 ± 7	14 ± 7	21 ± 6	13 ± 6	28 ± 26	13 ± 6	23 ± 3	15 ± 2	0.023
EPA (g)	0.1 ± 0.1	0.01 ± 0.01	0.2 ± 0.2	0.01 ± 0.01	0.1 ± 0.1	0.06 ± 0.09	0.1 ± 0.1	0.01 ± 0.01	0.1 ± 0.1	0.01 ± 0.01	0.1 ± 0.1	0.02 ± 0.03	0.08 ± 0.04	0.02 ± 0.02	0.019
DHA (g)	0.2 ± 0.2	0.03 ± 0.03	0.4 ± 0.4	0.02 ± 0.01	0.2 ± 0.3	0.12 ± 0.2	0.1 ± 0.1	0.02 ± 0.02	0.3 ± 0.3	0.02 ± 0.01	0.2 ± 0.2	0.05 ± 0.08	0.22 ± 0.09	0.04 ± 0.04	0.012
Fiber (g)	11 ± 6	30 ± 14	10 ± 5	31 ± 8	11 ± 6	30 ± 10	8 ± 3	30 ± 10	9 ± 6	29 ± 12	8 ± 6	29 ± 7	10 ± 2	30 ± 1	0.001
Sugar (g)	23 ± 7	138 ± 84	19 ± 8	129 ± 54	16 ± 4	125 ± 64	17 ± 8	136 ± 51	16 ± 6	132 ± 36	18 ± 7	135 ± 39	18 ± 2	132 ± 5	0.001

Values are mean ± SD (*n* = 7). P-values were obtained from pairwise t-tests between grand means. Determined from 3-day, 24-hour weighed dietary food records including 1 weekend day. LCHF, low carbohydrate high fat; HCLF, high carbohydrate low fat.

**Table 3 nutrients-14-01135-t003:** Changes in Plasma Lipids and Lipoproteins at Day 4, 14, 28, and 42 for LCHF and HCLF.

Variable	LCHF					HCLF					2 × 4 RM ANOVA (*p*-Value)		
	Day 4	Day 14	Day 28	Day 42	Mean	Day 4	Day 14	Day 28	Day 42	Mean	Condition	Time	Interaction
Total Cholesterol (mg/dL)	165.9 ± 21.7	185.6 ± 17.0	217.7 ± 16.6	220.3 ± 23.1	197.4 ± 26.3	154.7 ± 18.8	152.0 ± 29.5	152.7 ± 22.4	154.7 ± 19.4	153.4 ± 1.3	0.001	0.001	0.001
TC/HDL-C Ratio	2.5 ± 0.5	2.7 ± 0.6	3.0 ± 0.7	3.0 ± 0.7	2.8 ± 0.6	2.5 ± 0.5	2.6 ± 0.6	2.7 ± 0.6	2.8 ± 0.7	2.6 ± 0.6	0.035	0.001	0.011
HDL-C (mg/dL)	67.4 ± 15.5	67.7 ± 14.6	73.7 ± 18.9	73.6 ± 20.9	70.6 ± 3.5	63.4 ± 16.1	61.0 ± 18.5	59.0 ± 15.9	58.9 ± 16.2	60.6 ± 2.1	0.035	0.57	0.015
LDL-C (mg/dL)	84.7 ± 16.8	97.7 ± 20.4	122.0 ± 19.9	128.9 ± 17.7	108.3 ± 17.1	73.4 ± 14.2	71.9 ± 15.9	71.1 ± 17.5	77.4 ± 13.6	73.5 ± 12.8	0.001	0.001	0.001
Triglycerides (mg/dL)	69.1 ± 8.5	75.9 ± 6.1	79.1 ± 13.7	71.7 ± 10.4	73.9 ± 4.4	90.3 ± 22.2	96.9 ± 22.1	112.4 ± 30.5	88.6 ± 17.9	97.0 ± 10.9	0.005	0.13	0.44
VLDL (mg/dL)	13.9 ± 1.7	15.0 ± 1.2	15.9 ± 2.8	15.4 ± 3.3	15.0 ± 1.5	17.9 ± 4.5	19.4 ± 4.3	22.4 ± 6.2	17.9 ± 3.8	19.4 ± 2.8	0.004	0.23	0.28
TG/HDL-C Ratio	1.0 ± 0.3	1.2 ± 0.3	1.1 ± 0.4	1.0 ± 0.4	1.0 ± 0.1	1.5 ± 0.7	1.9 ± 0.8	2.1 ± 0.8	1.6 ± 0.6	1.8 ± 0.2	0.001	0.24	0.30
Glucose (mg/dL)	80.6 ± 3.3	85.7 ± 4.2	84.3 ± 7.1	82.6 ± 9.1	83.3 ± 4.0	89.3 ± 10.4	91.3 ± 12.5	86.0 ± 9.5	88.1 ± 9.4	88.7 ± 8.6	0.11	0.34	0.62
HbA1c (%)	4.9 ± 0.2	4.9 ± 0.2	4.9 ± 0.1	4.9 ± 0.1	4.9 ± 0.1	4.9 ± 0.2	4.9 ± 0.2	5.0 ± 0.3	5.0 ± 0.3	4.9 ± 0.1	0.82	0.49	0.26

To convert to SI units, multiply total cholesterol, LDL-C and HDL-C (mg/dL) × 0.0256 = mmol/L; multiply triglycerides (mg/dL) × 0.0113 =mmol/L. Values are Mean ± SD (*n* = 7). Bold face denotes statistical significance (*p* < 0.05). LCHF, low carbohydrate high fat; HCLF, high carbohydrate low fat; TC = total cholesterol; HDL-C = high density lipoprotein cholesterol; LDL-C = low density lipoprotein cholesterol; TG = triglycerides; VLDL = very low-density lipoprotein; HbA1c = glycated hemoglobin.

## Data Availability

The raw data that was analyzed for this manuscript will be made available and provided upon reasonable request.

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
