# Peer review of "The Effects of Carbohydrate versus Fat Restriction on Lipid Profiles in Highly Trained, Recreational Distance Runners: A Randomized, Cross-Over Trial"

_nutrients, 2022, doi:10.3390/nu14061135_

Round 1
Reviewer 1 Report
The authors evaluated changes in blood constituents during consumption of a high-carbohydrate diet and a low-carbohydrate, high-fat diet in a crossover design with seven subjects. The experiment was conducted properly and the results obtained are satisfactory.
Major comments
1) Washout is two weeks, but did macronutrient changes return to their original values? Figure 4 describes the amount of change from the starting value, but I would like to know the starting value for reference.
Minor comments
1) Line 22: The percentage of the HCLF diet totals 99.
2) Table 2: Did you use One-way to test all the data from Week 1 to 8 at the same time? At what weeks were there significant differences between LCHF and HCLF?
3) Please insert space after "<".
Author Response
Thank you to the reviewers for providing helpful input which we believe enhances the scientific quality of this manuscript. Please find our response to the reviewers below.
Reviewer #1
Major comments
1) Washout is two weeks, but did macronutrient changes return to their original values? Figure 4 describes the amount of change from the starting value, but I would like to know the starting value for reference.
Participants pre-trial dietary intake has been included for reference
“Participants’ pre-trial dietary intake averaged 3291 ± 348 Kcals and comprised 399 ± 69 g (51%) of carbohydrate, 141 ± 26 g (17%) of protein, and 129 ± 37 g (35%) of fat, consistent with the Standard American Diet (Shan et al., 2019).”
Shan Z, Rehm CD, Rogers G, Ruan M, Wang DD, Hu FB, Mozaffarian D, Zhang FF, Bhupathiraju SN. Trends in dietary carbohydrate, protein, and fat intake and diet quality among US adults, 1999–2016. JAMA. 2019;322:1178–87.
In the present study, a two-week washout period was used between the two 6-week dietary interventions, without feeding limitations. No dietary advice or recommendations were given to subjects during this period and subjects’ diet was not assessed.
We included this in the limitations section… “Additionally, a 2-week washout period may have been insufficient to eliminate carry-over effects of the previous dietary intervention. However, random allocation slightly reduced the influence of the short washout period.”
We, therefore, acknowledge this as a limitation to the study but do not believe it warrants rejection or non-publication of the paper as previous studies have also neglected to collect dietary data during the wash-out period.
Again, we would like to reiterate that the washout period used is consistent with that of previous studies (Zajac et al.,2014; Greene et al., 2018). In the study by Zajac et al. (2014), a one-week washout ‘recovery macrocycle’ was incorporated between the two 4-week dietary interventions. And in the study by Green et al. (2018) a two-week wash-out period was used between the two 12-week dietary intervention.
Minor comments
1) Line 22: The percentage of the HCLF diet totals 99.
Great observation – thank you for spotting this rounding error – here is the correction:
“…LCHF (6/69/25 %en carbohydrate/fat/protein) and HCLF (57/28/15 %en carbohydrate/fat/protein) …”
2) Table 2: Did you use One-way to test all the data from Week 1 to 8 at the same time? At what weeks were there significant differences between LCHF and HCLF?
The reviewer raises an interesting point that we omitted after statistical analysis. We decided that it would be best to conduct a pairwise t-test on the grand means to minimize the noise caused by weekly fluctuations and report a general diet-dependent effect. Secondly, we discovered that besides “energy intake” both LCHF and HCLF conditions differed significantly at all levels for all the variables reported. Indicating this effect with statistical symbols (e.g. *, #, etc.) turned out to clutter the limited table space and make the interpretation of results too redundant and cumbersome. Collectively, we updated the Table 2 caption to reflect the appropriate statistical tests that we conducted and provide additional detail for reader clarity.
3) Please insert space after "<".
All the “<” symbols are now followed by adequate space.
Reviewer 2 Report
This is a randomized cross-over trial of the effects of carbohydrate or fat restriction in recreational runners.
The study is well conducted, designed, and interesting. However, from an initial point of view, the authors do not make the rationale of the study clear. I suggest that the introduction be clearer about the importance of the study, based on rationale, especially on lipid profile outcomes.
Figure 2 shows only 6 individuals, and it was not clear why the seventh does not appear. Make it clear in the text.
In the methods, it is necessary to add the study power, based on the beta.
The study is confused about its purpose. They used good level runners but did not show the performance of runners after the interventions. It becomes interesting because just knowing the lipid profile after an intervention does not seem to produce relevant effects for readers. Please explain this point.
The conclusion is evasive and does not show a path or future perspective on the points studied.
Author Response
Thank you to the reviewers for providing helpful input which we believe enhances the scientific quality of this manuscript. Please find our response to the reviewers below.
Reviewer #2
The study is well conducted, designed, and interesting. However, from an initial point of view, the authors do not make the rationale of the study clear. I suggest that the introduction be clearer about the importance of the study, based on rationale, especially on lipid profile outcomes.
Figure 2 shows only 6 individuals, and it was not clear why the seventh does not appear. Make it clear in the text.
Thank you for bringing up this transparency issue. We followed up on your comment with a stronger explanation, at the end of the Introduction section, to highlight the novelty and importance of this study in exercise science and diet field.
Regarding Figure 2: The 12 bars shown in figure 2 report the weekly average nutrient intake of all 7 athletes combined. In other words, 6 bars are shown for each diet because each diet was 6 weeks long, so 7 athlete dietary records are compiled into one weekly bar. It is important that you brought this to our attention; we made appropriate changes to the figure caption to improve clarity.
In the methods, it is necessary to add the study power, based on the beta.
It seems there is concern about the small number of participants (n = 7), which raises questions about having a sufficient level of statistical power. Although there were attempts to increase the sample size, the experimental regimen required significant dietary restrictions, regular laboratory visits, and a restrictive inclusion criteria.
Other steps were taken to bolster the power of the study, including reducing the variance (by using only committed recreational runners), and using a crossover design. Indeed, the sample size was large enough to demonstrate dietary differences in carbohydrate, protein, and fat intake (both in grams and as a percentage), all p’s = 0.001 or smaller (see Table 2).
Also, there were significant differences on measures of triglycerides, LDL-C, HDL-C, total cholesterol, VLDL, and TG/HDL-C (see Tables 3). Hence, there was sufficient power to detect numerous differences between the two conditions.
See below for list of investigations with similar sample size:
Zajac et al., 2014; The effects of a ketogenic diet on exercise metabolism and physical performance in off-road cyclists; Crossover design; Subjects = 8
Phinney et al., 1983; The human metabolic response to chronic ketosis without caloric restriction: preservation of submaximal exercise capability with reduced carbohydrate oxidation; Crossover design; 5 subjects
Paoli et al., 2012; Ketogenic diet does not affect strength performance in elite artistic gymnasts; Crossover design; 8 subjects
Burke et al., 2000; Effect of fat adaptation and carbohydrate restoration on metabolism and performance during prolonged cycling; crossover design; 8 subjects
Carey et al., 2001; Effects of fat adaptation and carbohydrate restoration on prolonged endurance exercise; crossover design; 8 subjects
Lambert et al., 2001; High-Fat Diet Versus Habitual Diet Prior to Carbohydrate Loading: Effects on Exercise Metabolism and Cycling Performance; crossover design; 5 subjects
Shaw et al., 2019; Effect of a Ketogenic Diet on Submaximal Exercise Capacity and Efficiency in Runners; crossover design; 8 subjects
Heatherley et al., 2017; Effects of Ad libitum Low-Carbohydrate High-Fat Dieting in Middle-Age Male Runners; crossover design; 8 subjects
The purpose of the brief aforementioned review is to illustrate that many other similar studies have been published utilizing small sample sizes.
The study is confused about its purpose. They used good level runners but did not show the performance of runners after the interventions. It becomes interesting because just knowing the lipid profile after an intervention does not seem to produce relevant effects for readers. Please explain this point.
This is a good point – we hope that this question can be answered by our previous publication, conducted in the same runners, but focusing on running performance alone (Reference #15: Prins et. al., 2019; PMID: 31827359). For this paper, we wanted to focus solely on the diet related effects on blood lipids and briefly mention performance outcomes to minimize overlap with previously published data. We mention on lines 149-153 in Methods that prior statistical analyses revealed no change in running performance pre-, during, or post-experimental conditions. We included a statement that guides the reader to the paper aforementioned if they wish to learn more about performance metrics previously collected.
The conclusion is evasive and does not show a path or future perspective on the points studied.
The reviewer is correct: after re-reading the conclusion the take-home message is too ambiguous. We modified the conclusion to reflect appropriate future guidance based on what we reported. The current modification should make the final statement less evasive and guide future studies to further explore our primary outcomes. We hope that this text modification satisfies the expectations of future perspectives in a more precise manner.